# Hindering Factors and Perceived Needs for the Decision Making of Advanced Directives Among People with Dementia and Their Families

**DOI:** 10.3390/geriatrics10010019

**Published:** 2025-02-01

**Authors:** Hsiu-Ching Lin, Yu-Fang Lu, Ching-Hsueh Yeh, Jy-Jing Wang, Ya-Ping Yang

**Affiliations:** 1Department of Senior Citizen Services, National Tainan Junior College of Nursing, Tainan 700, Taiwan; othclin@ntin.edu.tw; 2Department of Nursing, Kaohsiung Medical University Hospital, School of Nursing, Kaohsiung Medical University, Kaohsiung 807, Taiwan; plk9751@gmail.com; 3Department of Medical Research, Kaohsiung Medical University Hospital, School of Nursing, Kaohsiung Medical University, Kaohsiung 807, Taiwan; chyeh@kmu.edu.tw; 4Department of Nursing, College of Medicine, National Cheng-Kung University, Tainan 701, Taiwan; ns127@mail.ncku.edu.tw; 5Department of Nursing, National Tainan Junior College of Nursing, Tainan 700, Taiwan

**Keywords:** dementia, end-of-life care, advanced directive, advance care planning, qualitative study, Taiwan

## Abstract

Making advanced directives is challenging in Asia. The hindering factors and perceived needs for advanced directives for people with dementia and their families have not been fully explored in Taiwan. In this study, we aimed to identify the barriers and perceived needs of people with mild dementia and the families of people with dementia within the cultural context of Taiwan for advanced directives. A qualitative descriptive design with purposive sampling and content analysis was used to collect and analyze the data. Thirteen people with mild dementia and thirty-two families of people with dementia were recruited. Our findings indicated that the hindering factors for people with mild dementia and the families of people with dementia to make advanced directives included “talking about death is a taboo”, “the timing is not right”, “cultural values of filial piety”, “male protagonist’s social status”, and “insufficient information on advanced directive”. The perceived needs for participants in making advanced directive decisions were “a wish to die without suffering”, “wanting to rely on others to make a decision”, and “an increased awareness of information”. This research offers valuable insights into the barriers and needs related to advanced directives for people with mild dementia and the families of people with dementia in Taiwan. These findings address the identified challenges and needs to develop effective solutions to help healthcare providers to better facilitate the decision-making process for advanced directives.

## 1. Introduction

Around 50 million people suffer from dementia, with nearly 10 million new cases being diagnosed every year worldwide. With the increase in the number of people with dementia (PWD), the total cost of dementia care has a great impact on the psychological, social, and economic aspects of society [1]. As dementia progresses, the ability to think and act decreases to the extent that PWD have difficulties in expressing their preferences with regard to their expectations for end-of-life care [2]. For lack of planning, end-of-life care for PWD depends on the decisions made by their families or healthcare agents, which may not be in agreement with their own decisions [3]. Therefore, an advanced directive (AD) is considered to be one of the important care issues for PWD at the end of their lives [4]. AD has received policy and legislative support in western countries [5]. In western countries, 8-20% of the general population sign AD documents [6]. However, it remains underdeveloped in most Asian societies [7,8]. For example, studies in Japan have reported that only 2% of AD documents are signed by PWD before the end of their lives, and more than 90% of those are signed by their families [9]. In Singapore, by 2017, around 10,000 advance care planning (ACP) conversations were had among the general population [8]. Until 2023, in Taiwan, less than 0.3% of adults had AD registration [10], and this was much lower for PWD.

Despite its importance being acknowledged, ACP is challenging in Asia [8]. Before having the legislative context of AD in Taiwan, people usually talk about end-of-life decisions regarding whether to make DNR orders. Fang et al. (2019) found that only 25% of families discuss DNR orders with PWD, and 32% of families assign the decision making to the doctor or other family members [11]. As dementia progresses, the ability to think and act decreases so that PWD cannot express their preferences about their expectations for end-of-life care [12]. In addition, in Asian countries, discussing death is a taboo subject, and many people believe that making an AD can bring about bad luck [8]. Initiating and carrying out the AD document signing continues to pose significant challenges [13]. General practitioners in the UK see dementia as a terminal disease and suggest that PWD should start to discuss AD when they are diagnosed with dementia [14]. If they can prepare AD documents or assign their healthcare agents in advance, healthcare providers can respect their choices and the ethical burden of making life-and-death decisions can be reduced to a certain extent [15].

For PWD in western countries, previous studies have indicated that the difficulties in signing DNR orders including delaying talking about the issue, depending on other family members to make the decision or an unwillingness to make a decision, lacking knowledge of AD, finding it difficult to discuss the matter, waiting to discuss the issue with a healthcare provider, being afraid to sign the documents, and being afraid of being given up [12]. For families of PWD in Western countries, difficulties include a lack of related knowledge, stress over making decisions or being unsure of their decision, anxiety accompanied by guilt, and being at a loss [12,16]. In regard to implementing AD for people in Asia, the general difficulties are in the rooted influence of Confucian cosmology, which makes people averse to discussing the mortality arrangement of the loved family member; in addition, healthcare professionals might rarely engage in an AD with their patients because they lack the knowledge and skill to make an AD and fear having conflicts with family members, as well as lacking a standard system for making an AD [8,13].

According to this systematic review, few studies on AD have taken Taiwanese culture into account, and the hindering factors and perceived needs for an AD for PWD and their families have not been well addressed [17]. In this study, we aimed to understand the hindering factors and the perceived needs of making an AD for this group of people within the Taiwanese context. We hope that our findings can be a reference to assist healthcare providers in facilitating the decision-making process of making an AD for PWD and their families and to motivate them to make end-of-life care decisions earlier, and thus benefit from having these documents on record.

## 2. Methods

### 2.1. Design and Data Collection

This study adopted a descriptive qualitative design, which enabled the elucidation of hindering factors and perceived needs for the decision making of an AD through the investigation of the perspectives of people with mild dementia and families with PWD [18]. People with mild dementia and their families in daycare centers and community care stations in southern Taiwan participated in semi-structured interviews. We aimed to recruit people with mild dementia because they already had a specific diagnosis of dementia, were capable of expressing their values and preferences, and had the need to know what might happen in the future for their end-of-life care. In addition, we recruited families with PWD to know their hindering factors and perceived needs in the decision making of end-of-life care for PWD. Thus, purposive sampling was adopted. The inclusion criteria for the people with mild dementia were the following: (1) for those with an education level of senior high school and above, a Mini-Mental State Examination (MMSE) score <24; (2) for those with an education level below senior high school, a MMSE score <18; and (3) those able to speak Mandarin or Taiwanese. The inclusion criteria for the families of PWD were the following: (1) those who had taken care of a person with dementia for more than 6 months and (2) those able to speak Mandarin or Taiwanese.

Before conducting the study, two interviewers were trained in interview techniques and the consistency of data collection. These interviewers were required to be university graduates in nursing with over five years of experience in dementia care. The purpose and process of the interview were thoroughly explained to the participants beforehand, and informed consent was obtained prior to both the interview and the recording.

Each interview lasted 20–30 min and consisted of open-ended questions. After each interview, one of the authors transcribed the interviews verbatim. When all interviews had been preliminarily analyzed, the research team convened to ensure that no new patterns emerged and that thematic saturation was reached [19]. The final sample comprised 13 people with mild dementia and 32 families of people with dementia.

### 2.2. Instruments

The semi-structured interviews included two parts: (1) basic demographic information of the people with mild dementia and families of PWD including whether they have an AD decision and (2) an interview guide was designed to understand their hindering factors and perceived needs for an AD, including 1. “What factor do you think to hinder your decision on AD?” and 2. “What kinds of assistance do you need for making an AD decision?”

### 2.3. Ethical Approval

The Institutional Review Board of Kaohsiung Medical University Hospital (IRB number: KMUHIRB-SV(I)-20170059) approved this study. People with mild dementia and families of PWD signed informed consent forms, and then the interviews were conducted. The participants could stop the interview at any time if they felt uncomfortable.

### 2.4. Data Analysis

Content analysis was employed to analyze the interview transcripts. Following Doyle et al. (2020) and Lindgren et al. (2020), the analysis process included data management, repeated reading, condensation of words and subcategories, identification of themes, reflection, and achieving data saturation [18,19,20,21]. The procedure was as follows: First, the interviewers transcribed the interviews. Two researchers then independently read and reflected on the transcripts. Subsequently, they collaborated to develop a consensus on the coding scheme. Data were coded and categorized. Any discrepancies were discussed with the research group to reach an agreement on the main themes, defined using the categories and codes. After abstracting the data, the concepts emerged, and the themes were linked to the research questions and described in detail [21].

Respondent validation was one of the verification procedures and was undertaken in this study to seek the participants’ judgments of the accuracy of the interpretations and findings [22]. The analyses were discussed between the first author and the corresponding author back and forth to ensure that the methodological and analytic decisions could assist in the development of dependability in qualitative research. All authors arrived at a consensus regarding the condensed meaningful chunks, subthemes, and themes through triangulation to maintain methodological rigor [22].

## 3. Results

### 3.1. Participants’ Characteristics

This study recruited two groups of participants. The first group comprised 13 people with mild dementia and an mean age of 81 (range: 66–94), and P1–P13 was used to represent them. The second group comprised 32 family members of PWD with an mean age of 65 (range: 40–90), and F1–F32 was used to represent them (Table 1).

For experience with ADs, the results showed that, among 13 people with mild dementia, more than 80% did not sign DNR orders, an agreement to not receive life-sustaining treatment, a letter of attorney for their healthcare agent, or a letter of intent for hospice palliative care. Less than a quarter had discussed an AD with their families. For 32 families of PWD, as far as the experiences and perceived needs of making an AD were concerned, the results showed that more than 80% of the family members did not sign AD documents for PWD, and approximately 70% had not discussed an AD with PWD. Concerning with whom to discuss an AD, 10 to 20% would discuss it with their families and PWD, but less than 10% would discuss it with healthcare providers (Table 2).

### 3.2. Hindering Factors

Hindering factors are factors preventing people with mild dementia and families of PWD in making an AD. Five themes for this were identified: talking about death is a taboo, the timing is not right, the male protagonist’s social status, cultural values of filial piety, and insufficient information on ADs (Table 3). These themes are explained below.

#### 3.2.1. Theme 1–Talking About Death Is a Taboo

Many participants mentioned that death or dying was scary, and few people wanted to take the initiative to talk about it. Therefore, they avoided talking about issues related to death or dying such as a DNR, which led them not to sign the DNR.

“*I’m afraid of discussing the issue of death.*”(P7)

“*I haven’t discussed it with her because I’m afraid it’s a taboo subject for her.*”(F25)

“*When I think about DNR orders, I feel worried so I don’t want to think or talk about it.*”(P7)

“*If we talked about it (DNR decision), it would make me sad.*”(F10)

#### 3.2.2. Theme 2–The Timing Is Not Right

This theme refers to the fact that either people with mild dementia or families of PWD thought the disease course of dementia was still fine and tolerable, so it was not the time to discuss the issue of an AD with others. However, when the disease became unstable or worsened, they started to consider the issue of an AD.

“*I am still healthy now, so I don’t think about it too much.*”(P6)

“*Because it hasn’t happened yet, I never think about it.*”(P8)

“*It’s not the time yet so we don’t think about it.*”(F14)

“*When he gets worse, he will be admitted to an ICU. Then I will consider it.*”(F1)

#### 3.2.3. Theme 3–Cultural Values of Filial Piety

Cultural values of filial piety refer to the children of PWD who were expected to be filial to their parents. While their sick parent expected their children to make an AD on their behalf, the children were still hesitant to make the decision too early, fearing that it might be perceived as unfilial.

“*When the time for decision making comes, my children are filial so they will help me to make a decision.*”(P7)

“*If I sign an AD, my relatives will think I am unfilial.*”(F20)

#### 3.2.4. Theme 4–Male Protagonist’s Social Status

When the time for making an end-of-life decision came, participants with mild dementia may prefer their families to do so. Especially, they preferred males such as sons to be the decision-maker over females.

“*Let his son decide.*”(P19)

“*Because I am illiterate, I won’t make any decision, but let my son do so instead.*”(P9)

“*Because I am a daughter-in-law, I cannot make the decision myself. She (her mother-in-law) has her own concerns. It’s better to let her son ask.*”(F10)

#### 3.2.5. Theme 5–Insufficient Information on ADs

Many participants expressed that they did not know what a DNR or AD was, nor how to sign for it.

“*I didn’t know we can sign the Letter of Intent first, so I did not help him/her to do so.*”(P2)

“*My family with dementia has told us not to resuscitate, but I don’t know how to sign the agreement.*”(P16)

“*I didn’t know how to gain the information.*”(F8)

“*I need someone to tell me how to make AD.*”(F8)

### 3.3. Perceived Needs

Perceived needs are what people with mild dementia and families of PWD feel they need when making an AD decision. Three themes emerged as follows: “a wish to die without suffering”, “wanting to rely on others to make a decision”, and “increased awareness of information” (Table 3).

#### 3.3.1. Theme 1–A Wish to Die Without Suffering

When participants made AD decisions, they thought that such a decision would allow their loved ones or themselves to die without suffering.

“*I have already made it clear to my daughter that I don’t need to be resuscitated because I heard my relatives and friends say it’s painful.*”(P3)

“*I don’t want him suffering, and I prefer he has a good death.*”(F32)

“*I wish him to die without suffering.*”(F8)

“*Making an intubation cannot change anything, so I hope him to have a natural death.*”(F28)

#### 3.3.2. Theme 2–Wanting to Rely on Others to Make a Decision

When the time came to make an end-of-life decision for PWD, participants wanted to count on others such as their children or healthcare providers to make this decision for them.

“*When making a decision is necessary, my children are filial to me and they will do this for me.*”(P7)

“*Let my kids make decisions for me.*”(P8, P10)

“*Let the doctor make a decision (for PWMD).*”(F11)

#### 3.3.3. Theme 3–An Increased Awareness of Information

Because of the insufficient information on ADs for participants, they looked forward to raising their awareness of ADs. The subthemes of this include “gaining information from healthcare providers”, “needing an AD kit”, and “publicly promoted information”.

##### Subtheme 1–Gaining Information from Healthcare Providers

Many participants expressed a need for the healthcare providers including nurses and physicians to actively provide information related to ADs or to discuss it to them.

“*It hasn’t happened yet, so can the healthcare providers give me sufficient information?*”(P1)

“*Explanations by healthcare providers and communication with my family are needed.*”(P8)

“*When my situation changes in the hospital, if the doctor can talk and discuss this with me, it would be helpful.*”(P12)

“*The explanation about AD from healthcare providers is needed.*”(F1)

“*I didn’t know how to gain the information, because it was not provided actively by the healthcare providers.*”(F8)

##### Subtheme 2–Needing an AD Kit

Participants pointed out that service providers could make an AD kit available and easily understandable for them.

“*Easy to understand clips for AD.*”(F5)

“*A handbook of AD available.*”(F5)

“*Documents for healthcare providers explaining AD clearly.*”(F3)

“*Available forums on AD.*”(F7)

##### Subtheme 3–Publicly Promoted Information

The participants pointed out the need for the information about ADs to be publicly promoted to gain wider attention.

“*AD can be promoted via TV.*”(F16)

“*AD can be explained on TV, news or broadcasts.*”(F22)

“*The hospital can provide an AD website.*”(F15)

“*More AD lectures can be held in churches.*”(F7)

## 4. Discussion

This study explored the experiences and perceived needs for an AD among people with mild dementia and the families of PWD. Among the participants, only 2 people with mild dementia and fewer than 10 family members of PWD had experience in making AD documents. This is similar to the findings of a previous study, indicating their low intention to discuss this topic [11].

The finding in which the participants expressed avoiding discussing death is consistent with the conservative attitude toward discussing death with others among Taiwanese people [23]. Due to the rooted cultural influence of Confucianism in Asia, people might see death as a taboo [7,8,22,23,24]. It also suggests that participants were generally satisfied with the stable dementia status of themselves and their family members, leading them to avoid discussing the issue of death. Because of the culturally bound fear of dying sooner, PWD could not think about what their expectations and preferences in the last stage of their lives were, and they preferred to let their families decide for them instead [2,16,21,23]. However, in Asian tradition, if children discuss the issue of death with their parents, it may be seen as unfilial and disrespectful behavior [8,23,25,26]. Thus, families of PWD have difficulties in discussing end of life decisions with them earlier [11,23]. This corresponds to the fact that families of PWD are reluctant and feel guilty about discussing ADs with them due to the feelings of uncertainty related to dementia’s course and the cultural influences of death aversion and filial piety [3,8,11,23].

The timing of discussing an AD typically starts when the disease reaches an unstable stage. Many family members with PWD in this study refused to initiate the discussion of end-of-life care because they felt it was not the right time. Some participants with mild dementia wanted to delay the conversation because they still looked fine and stable. These results are consistent with Lee’s studies [23,27,28]. Previous studies have reported that the timing of the discussion of end-of-life decisions might simply be initiated and repeated among PWD, their family carers, and healthcare providers when the balance between the disease course of dementia and decision-making capacity collapses [17,29,30]. It is highly recommended to initiate the discussion of ACP at an earlier stage and revise it as needed because the cognitive decline of PWD is mild. This timing allows them to actively engage with their families and healthcare professionals [17]. Recognizing that patients’ needs and wishes can change over time, ACP facilitators encourage ongoing conversations [8].

The decision-makers for people with mild dementia were usually their family members, and sons were especially favored. Due to the strong adherence to filial duty, sons are traditionally expected to reciprocate care and upbringing in such a way [31]. Thus, they are preferred as the decision-makers for their parents with dementia over daughters. An empirical study in Taiwan found that sons are preferred surrogates, ranking higher than daughters and followed by the spouse [24]. Daughters-in-law, being non-biological relatives, often play the role of caregivers [32].

In this study, most individuals with mild dementia and their families were unfamiliar with ADs and had not received information about them. The lack of available information and the unclear medical status of PWD pose challenges to raising public awareness about ADs. To address this, educational materials should be made widely accessible through various multimedia channels.

In Taiwan, common information sources for the elderly include TV, newspapers, magazines, oral communication from friends and neighbors, professional healthcare providers, and the Internet. However, people with mild dementia often engage in activities such as watching TV and local drama series, which limits their exposure to AD information. This insufficient information dissemination impedes their involvement in ACP. Families of PWD recommended that AD information be publicly promoted and that an AD kit be made available. As emphasized by Ho et al. [8], sustainable implementation of ACP requires a public health strategy involving all healthcare professionals and societal members, supported by collective social action and government leadership. Resources supporting ACP can facilitate quality conversations about ADs. Participants suggested that TV, community-based programs, handbooks, healthcare education, and promotional videos are effective channels for disseminating this information.

“A wish to die without suffering” expressed by our participants indicated that either themselves or their family members wish to die without pain. Since an AD includes the decision of accepting or refusing life-sustaining treatment, artificial nutrition, and hydration, or other medical care, people should know these options for their desires for a good death. A good death is a central concern of palliative care. When they decide on their wishes for a natural death, they can adjust their mood and face the issue of death with more ease [13].

When the disease progresses, PWD and their families have no choice but to discuss it [17,33]. Our study found that participants tended to rely on their children or healthcare providers to make the medical decisions for them. This shows that Taiwanese people are likely to discuss and make decisions within their own family, as collective decision making has been dominant in Asia for some time [8,9,11,25,34]. East Asians, such as Taiwanese people, often see healthcare providers as figures of authority and tend to play a passive role in seeking advice from them [35]. This also explains why people with dementia hoped that the healthcare team could provide information and that their families could know how to make an AD for them to minimize family burden and enhance appreciation for their care preferences [34,36]. Ho et al. [8] suggested that a family-centered ACP in Asia can have greater recognition than the Western notions of autonomy and patient-centered care.

During the interview process, the families of PWD expected more active involvement from the healthcare team [8]. However, healthcare providers in Asia might seldom actively provide information because of the unpredictable nature of the disease, the fear of causing stress to PWD and their families, uncertainty about the primary initiator of ACP discussions, or a lack of confidence in this [9,11,13,17,30].

Recent advice to healthcare providers also indicates that early AD interventions are valuable for PWD and their families, such as in understanding the meaning of suffering to them so they are the most suitable party to initiate the conversation; facilitating ACP is related to the time they spend on individuals [3,17,37]. Thus, they need to have sufficient knowledge and skills to be able to guide PWD and their families [13]. The education including disease course, death, dying, bereavement, hospice, palliative care, ACP, and communications skills, and with the support of ongoing institutional and community-based research, will empower healthcare providers to become more engaged in ADs for PWD and their families, and lead to further advocacy and social change [8,17]. It is suggested that the success of the healthcare providers’ intervention and sustainability relies on pre-implementation preparation into routine care through training, understanding, and the codesigning of existing workplace systems/workflow adaptions to workplace practices with support from relevant stakeholders [38].

This study’s findings may not apply to people with mild dementia and families of PWD who are well informed about ADs. Finally, this study reflects parts of Taiwanese culture, such as the priority of filial piety, the taboo subject of death, and the male protagonist’s social status, which may not apply to other cultures.

## 5. Conclusions

Making an AD has become a crucial aspect of end-of-life quality care in many advanced societies, but the official registered rate remains relatively low in some parts of Asia [8,9,10]. This research sheds light on the hindering factors and perceived needs related to making an AD for individuals with mild dementia and their families in this region. Healthcare providers can leverage these insights to better support the decision-making process, address challenges, and develop effective solutions. Future research will focus on creating patient decision aids to facilitate informed decision making, aiming to build healthcare providers’ confidence in initiating early AD discussions and fostering greater engagement between patients and their families.

## Figures and Tables

**Table 1 geriatrics-10-00019-t001:** Demographic characteristics of people with mild dementia and families of PWD.

		People with Mild Dementia (n = 13)	Families of PWD (n = 32)
Age		81.7 (7.6) *	65.0 (11.0) *
Gender	Male	3 (23.1)	11 (34.4)
	Female	10 (76.9)	21 (65.6)
Relationship with people with dementia	Spouse	–	7 (21.9)
	Direct relative	–	16 (50.0)
	Relation by marriage	–	9 (28.1)
Marriage status	Single	0 (0.0)	3 (9.4)
	Married	3 (23.1)	28 (87.5)
	Widowed	10 (76.9)	1 (3.1)
Religion	None	1 (7.7)	3 (9.4)
	General folk belief	8 (61.5)	15 (46.9)
	Buddhism	3 (23.1)	11 (34.4)
	Catholicism	1 (7.7)	3 (9.4)
Education level	No education	5 (38.5)	1 (3.1)
	Below junior high school	8 (61.5)	10 (31.3)
	Senior and vocational high school	0 (0.0)	21 (65.6)
Who paid the medical expenses	Shared by children	13 (100)	18 (56.3)
	People with dementia or paid by the spouse	0 (0.0)	14 (43.8)
Monthly income of the families of people with dementia (USD)	No income	–	10 (31.3)
	<USD 1000		9 (28.1)
	≧USD 1000		13 (40.6)
Subsidies or insurance	None	–	25 (78.1)
	Yes	–	7 (21.9)

Data are presented as frequency (percentage) or mean (standard deviation) *; –: not applicable.

**Table 2 geriatrics-10-00019-t002:** Experience with ADs of people with mild dementia and families of PWD.

		People with Mild Dementia (n = 13)	Families of PWD (n = 32)
**Agreement of DNR in the final stage of life**			
Sign/Sign for PWD	Yes	2 (15.4)	5 (15.6)
Discuss/Discuss with PWD	Yes	3 (23.1)	10 (31.3)
Discussed with whom	Family	3 (23.1)	11 (34.4)
	Healthcare providers	0 (0.0)	3 (9.4)
	PWD	–	10 (31.3)
**Agreement of not receiving life-sustaining treatment**			
Sign/Sign for PWD	Yes	1 (7.7)	3 (9.4)
Discuss/Discuss with PWD	Yes	1 (7.7)	5 (15.6)
Discussed with whom	Family	1 (7.7)	6 (18.8)
	Healthcare providers	0 (0.0)	1 (3.1)
	PWD	–	5 (15.6)
**Letters of attorney for healthcare agents**			
Sign/Sign for PWD	Yes	0 (0.0)	2 (6.3)
Discuss/Discuss with PWD	Yes	2 (15.4)	2 (6.3)
Discussed with whom	Family	2 (15.4)	2 (6.3)
	Healthcare providers	0 (0.0)	1 (3.1)
	PWD	–	2 (6.3)
**Letter of intent for advanced hospice palliative care**			
Sign/Sign for PWD	Yes	1 (7.7)	1 (3.1)
Discuss/Discuss with PWD	Yes	2 (15.4)	5 (15.6)
Discussed with whom	Family	2 (15.4)	4 (12.5)
	Healthcare providers	0 (0.0)	0 (0.0)
	PWD	–	5 (15.6)

Data are presented as frequency (percentage); –: not applicable.

**Table 3 geriatrics-10-00019-t003:** Hindering factors and perceived needs for the decision making on advanced directives among people with mild dementia and families of people with dementia.

Concept	Theme	Subtheme
Hindering factors	Talking about death is a taboo	
	The timing is not right	
	Cultural values of filial piety	
	Male protagonist’s social status	
	Insufficient information on ADs	
Perceived needs	A wish to die without suffering	
	Wanting to rely on others to make a decision	
	An increased awareness of information	Gaining information from healthcare providers
		Needing an AD kit
		Publicly promoted information

## Data Availability

Data cannot be publicly shared due to ethical restrictions. Researchers who meet the criteria for accessing confidential data can obtain it from the Institutional Data Access/Institutional Review Board of Kaohsiung Medical University Hospital by contacting irb@kmuh.org.tw.

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
