# Peer review of "Hindering Factors and Perceived Needs for the Decision Making of Advanced Directives Among People with Dementia and Their Families"

_geriatrics, 2025, doi:10.3390/geriatrics10010019_

Round 1
Reviewer 1 Report
Comments and Suggestions for Authors
Dear Authors
First, I would like to congratulate you on the topic you approached. It is an essential issue in the care of the patients with dementia.
After reading your submission, I think there are some issues that you need to correct to improve it before it is ready for publication.
Please understand that my comments are aimed at helping you enhance your manuscript for publication.
So:
1) Introduction: good introduction; diversity of the references, as well as their relevance.
2) Methods:
a) Design and data collection: you should mention where the subjects/patients and families were recruited;
b) Instruments: you must present how you built the interview script and how it was validated;
3) Results:
a) The data presentation can be improved. It isn't evident. You should present the data referred to in Table 1, and only then should you go to the data in Table 2.
b) Table 1: you used the mean and deviation standard to present the age but did not analyse the normality. The distribution is probably non-normal with so few subjects, so the median and the interquartile range are the best measures.
c) Table 1 & 2: you mention in the footnote that the data are presented as frequency (percentage). However, the percentages are not presented in the table. Please revise;
d) Table 2; you mention in the footnote that the data are presented as mean (standard deviation). However, you have no continuous variable for using these measures in the table. Please revise;
e) Lines 171-173: the source of the assumptions mentioned is not clear. Please check and revise;
4) Discussion: you must present the work's limitations, weaknesses and bias.
5) Conclusions: you should present the practice's strengths, recommendations and implications.
Kind regards
Author Response
|
1. Point-by-point response to Comments and Suggestions for Authors |
|
Comments 1: Design and data collection: you should mention where the subjects/patients and families were recruited. |
|
Response 1: Thank you for pointing this out. Please see lines 90-91 for the information. |
|
Comments 2: Instruments: you must present how you built the interview script and how it was validated. Response 2: Thank you for pointing this out. Please see lines 129-143 for the information. |
|
Comments 3: The data presentation can be improved. It isn't evident. You should present the data referred to in Table 1, and only then should you go to the data in Table 2. Response 3: Thank you for pointing this out. Please see lines 146-161. Comments 4 Table 1: you used the mean and deviation standard to present the age but did not analyse the normality. The distribution is probably non-normal with so few subjects, so the median and the interquartile range are the best measures. Response 4: Thank you for pointing this out. Please see lines146-148. Comments 5 Table 1 & 2: you mention in the footnote that the data are presented as frequency (percentage). However, the percentages are not presented in the table. Please revise. Response 5: Thank you for pointing this out. Please see the revised Table 1 and Table 2. Comments 6 Table 2; you mention in the footnote that the data are presented as mean (standard deviation). However, you have no continuous variable for using these measures in the table. Please revise. Response 6: Thank you for pointing this out. Please see the revised Table 2. Comments 7 Lines 171-173: the source of the assumptions mentioned is not clear. Please check and revise Response 7: Thank you for pointing this out. Please see the revised Table 2 and lines 153-161. Comments 8 Discussion: you must present the work's limitations, weaknesses and bias. Response 8: Thank you for pointing this out. Please see lines 359-362 for the information. Comments 9 Conclusions: you should present the practice's strengths, recommendations and implications. Response 9: Thank you for pointing this out. Please see lines 366-372 for the information.
|
Reviewer 2 Report
Comments and Suggestions for Authors
Thank you for your work on advance directives for people living with dementia, in Taiwan. I have a small number of minor suggestions for you to consider and correct.
Line 155 - Check that tables are fully visible using landscape or otherwise
Line 179 - The table shows no subthemes for the category of meaning on hindering the implementation of AD. Check formatting or otherwise consider what coding was drawn from the data. I note the expression of fear in some quotes and this could be an important code (or sub theme).
Line 277 - Add a little more explanation of "death aversion". Add any citation to the literature on this- either a basic human concept, or is it culturally bound?
Line 365 - I acknowledge that official reporting may be very accurate in your setting. Please comment, as (in my experience) it is not always reliable
Comments on the Quality of English Language
Line 72 - For clarification, adjust the wording of "regarding to"
Line 157 - fix spelling "applicab"
Line 344 - Refine the word "unclarity"
Author Response
|
1. Point-by-point response to Comments and Suggestions for Authors |
||||
|
Round 2
Reviewer 1 Report
Comments and Suggestions for Authors
Dear Authors
You considered my comments and suggestions, so the submission is ready to publish.
Kind regards